# Factoring Variations in Natural Images with Deep Gaussian Mixture Models

**Aäron van den Oord, Benjamin Schrauwen**
Electronics and Information Systems department (ELIS), Ghent University
{aaron.vandenoord, benjamin.schrauwen}@ugent.be

## Abstract

Generative models can be seen as the swiss army knives of machine learning, as many problems can be written probabilistically in terms of the distribution of the data, including prediction, reconstruction, imputation and simulation. One of the most promising directions for unsupervised learning may lie in Deep Learning methods, given their success in supervised learning. However, one of the current problems with deep unsupervised learning methods, is that they often are harder to scale. As a result there are some easier, more scalable shallow methods, such as the Gaussian Mixture Model and the Student-t Mixture Model, that remain surprisingly competitive. In this paper we propose a new *scalable* deep generative model for images, called the Deep Gaussian Mixture Model, that is a straightforward but powerful generalization of GMMs to multiple layers. The parametrization of a Deep GMM allows it to efficiently capture products of variations in natural images. We propose a new EM-based algorithm that scales well to large datasets, and we show that both the Expectation and the Maximization steps can easily be distributed over multiple machines. In our density estimation experiments we show that deeper GMM architectures generalize better than more shallow ones, with results in the same ballpark as the state of the art.

## 1 Introduction

There has been an increasing interest in generative models for unsupervised learning, with many applications in Image processing [1, 2], natural language processing [3, 4], vision [5] and audio [6]. Generative models can be seen as the swiss army knives of machine learning, as many problems can be written probabilistically in terms of the distribution of the data, including prediction, reconstruction, imputation and simulation. One of the most promising directions for unsupervised learning may lie in Deep Learning methods, given their recent results in supervised learning [7]. Although not a universal recipe for success, the merits of deep learning are well-established [8]. Because of their multilayered nature, these methods provide ways to efficiently represent increasingly complex relationships as the number of layers increases. "Shallow" methods will often require a very large number of units to represent the same functions, and may therefore overfit more.

Looking at real-valued data, one of the current problems with deep unsupervised learning methods, is that they are often hard to scale to large datasets. This is especially a problem for unsupervised learning, because there is usually a lot of data available, as it does not have to be labeled (e.g. images, videos, text). As a result there are some easier, more scalable shallow methods, such as the Gaussian Mixture Model (GMM) and the Student-t Mixture Model (STM), that remain surprisingly competitive [2]. Of course, the disadvantage of these mixture models is that they have less representational power than deep models.

In this paper we propose a new *scalable* deep generative model for images, called the Deep Gaussian Mixture Model (Deep GMM). The Deep GMM is a straightforward but powerful generalization of Gaussian Mixture Models to multiple layers. It is constructed by stacking multiple GMM-layers on

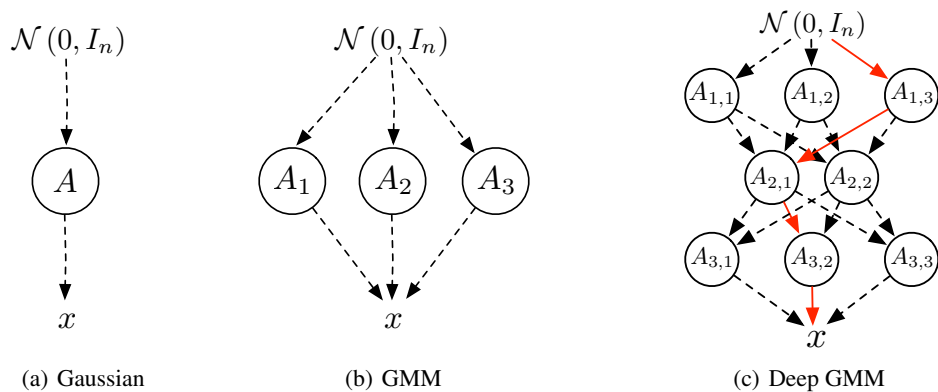

(a) Gaussian       (b) GMM       (c) Deep GMM

Figure 1: Visualizations of a Gaussian, GMM and Deep GMM distribution. Note that these are not graphical models. This visualization describes the connectivity of the linear transformations that make up the multimodal structure of a deep GMM. The sampling process for the deep GMM is shown in red. Every time a sample is drawn, it is first drawn from a standard normal distribution and then transformed with all the transformations on a randomly sampled path. In the example it is first transformed with $A_{1,3}$, then with $A_{2,1}$ and finally with $A_{3,2}$. Every path results in differently correlated normal random variables. The deep GMM shown has $3 \cdot 2 \cdot 3 = 18$ possible paths. For each square transformation matrix $A_{i,j}$ there is a corresponding bias term $b_{i,j}$ (not shown here).

top of each other, which is similar to many other Deep Learning techniques. Although for every deep GMM, one could construct a shallow GMM with the same density function, it would require an exponential number of mixture components to do so.

The multilayer architecture of the Deep GMM gives rise to a specific kind of parameter tying. The parameterization is most interpretable in the case of images: the layers in the architecture are able to efficiently factorize the different variations that are present in natural images: changes in brightness, contrast, color and even translations or rotations of the objects in the image. Because each of these variations will affect the image separately, a traditional mixture model would need an exponential number of components to model each combination of variations, whereas a Deep GMM can factor these variations and model them individually.

The proposed training algorithm for the Deep GMM is based on the most popular principle for training GMMs: Expectation Maximization (EM). Although stochastic gradient (SGD) is also a possible option, we suggest the use of EM, as it is inherently more parallelizable. As we will show later, both the Expectation and the Maximization steps can easily be distributed on multiple computation units or machines, with only limited communication between compute nodes. Although there has been a lot of effort in scaling up SGD for deep networks [9], the Deep GMM is parallelizable by design.

The remainder of this paper is organized as follows. We start by introducing the design of deep GMMs before explaining the EM algorithm for training them. Next, we discuss the experiments where we examine the density estimation performance of the deep GMM, as a function of the number of layers, and in comparison with other methods. We conclude in Section 5, where also discuss some unsolved problems for future work.

## 2  Stacking Gaussian Mixture layers

Deep GMMs are best introduced by looking at some special cases: the multivariate normal distribution and the Gaussian Mixture model.

One way to define a multivariate normal variable x is as a standard normal variable $z \sim \mathcal{N}\left(0, I_n\right)$ that has been transformed with a certain linear transformation: $x = Az + b$, so that

$$p\left(x\right) = \mathcal{N}\left(x|b, AA^T\right).$$

This is visualized in Figure 1(a). The same interpretation can be applied to Gaussian Mixture Models, see Figure 1(b). A transformation is chosen from set of (square) transformations $A_i, i = 1 \ldots N$ (each having a bias term $b_i$) with probabilities $\pi_i, i = 1 \ldots N$, such that the resulting distribution becomes:

$$p(x) = \sum_{i=1}^{N} \pi_i \mathcal{N}\left(x|b_i, A_i A_i^T\right).$$

With this in mind, it is easy to generalize GMMs in a multi-layered fashion. Instead of sampling one transformation from a set, we can sample a path of transformations in a network of $k$ layers, see Figure 1(c). The standard normal variable $z$ is now successively transformed with a transformation from each layer of the network. Let $\Phi$ be the set of all possible paths through the network. Each path $p = (p_1, p_2, \ldots, p_k) \in \Phi$ has a probability $\pi_p$ of being sampled, with

$$\sum_{p \in \Phi} \pi_p = \sum_{p_1, p_2, \ldots, p_k} \pi_{(p_1, p_2, \ldots, p_k)} = 1.$$

Here $N_j$ is the number of components in layer $j$. The density function of x is:

$$p(x) = \sum_{p \in \Phi} \pi_p \mathcal{N}\left(x|\beta_p, \Omega_p \Omega_p^T\right), \tag{1}$$

with

$$\beta_p = b_{k,p_k} + A_{k,i_k}\left(\ldots\left(b_{2,p_2} + A_{2,p_2} b_{1,p_1}\right)\right) \tag{2}$$

$$\Omega_p = \prod_{j=k}^{1} A_{j,p_j}. \tag{3}$$

Here $A_{m,n}$ and $b_{m,n}$ are the $n$'th transformation matrix and bias of the $m$'th layer. Notice that one can also factorize $\pi_p$ as follows: $\pi_{(p_1, p_2, \ldots, p_k)} = \pi_{p_1} \pi_{p_2} \ldots \pi_{p_k}$, so that each layer has its own set of parameters associated with it. In our experiments, however, this had very little difference on the log likelihood. This would mainly be useful for very large networks.

The GMM is a special case of the deep GMM having only one layer. Moreover, each deep GMM can be constructed by a GMM with $\prod_j^k N_j$ components, where every path in the network represents one component in the GMM. The parameters of these components are tied to each other in the way the deep GMM is defined. Because of this tying, the number of parameters to train is proportional to $\sum_j^k N_j$. Still, the density estimator is quite expressive as it can represent a large number of Gaussian mixture components. This is often the case with deep learning methods: Shallow architectures can often theoretically learn the same functions, but will require a much larger number of parameters [8]. When the kind of compound functions that a deep learning method is able to model are appropriate for the type of data, their performance will often be better than their shallow equivalents, because of the smaller risk of overfitting.

In the case of images, but also for other types of data, we can imagine why this network structure might be useful. A lot of images share the same variations such as rotations, translations, brightness changes, etc.. These deformations can be represented by a linear transformation in the pixel space. When learning a deep GMM, the model may pick up on these variations in the data that are shared amongst images by factoring and describing them with the transformations in the network.

The hypothesis of this paper is that Deep GMMs overfit less than normal GMMs as the complexity of their density functions increase because the parameter tying of the Deep GMM will force it to learn more useful functions. Note that this is one of the reasons why other deep learning methods are so successful. The only difference is that the parameter tying in deep GMMs is more explicit and interpretable.

A closely related method is the deep mixture of factor analyzers (DMFA) model [10], which is an extension of the Mixture of Factor Analyzers (MFA) model [11]. The DMFA model has a tree structure in which every node is a factor analyzer that inherits the low-dimensional latent factors

from its parent. Training is performed layer by layer, where the dataset is hierarchically clustered and the children of each node are trained as a MFA on a different subset of the data using the MFA EM algorithm. The parents nodes are kept constant when training its children. The main difference with the proposed method is that in the Deep GMM the nodes of each layer are connected to all nodes of the layer above. The layers are trained jointly and the higher level nodes will adapt to the lower level nodes.

## 3    Training deep GMMs with EM

The algorithm we propose for training Deep GMMs is based on Expectation Maximization (EM). The optimization is similar to that of a GMM: in the E-step we will compute the posterior probabilities $\gamma_{np}$ that a path $p$ was responsible for generating $x_n$, also called the *responsibilities*. In the maximization step, the parameters of the model will be optimized given those responsibilities.

### 3.1    Expectation

From Equation 1 we get the the log-likelihood given the data:

$$\sum_n \log p\left(x_n\right) = \sum_n \log \left[ \sum_{p \in \Phi} \pi_p \mathcal{N}\left(x_n | \beta_p, \Omega_p \Omega_p^T\right) \right].$$

This is the global objective for the Deep GMM to optimize. When taking the derivative with respect to a parameter $\theta$ we get:

$$\begin{aligned}
\nabla_\theta \sum_n \log p\left(x_n\right) &= \sum_{n,p} \frac{\pi_p \mathcal{N}\left(x_n | \beta_p, \Omega_p \Omega_p^T\right) \left[\nabla_\theta \log \mathcal{N}\left(x_n | \beta_p, \Omega_p \Omega_p^T\right)\right]}{\sum_q \pi_q \mathcal{N}\left(x_n | \beta_q, \Omega_q \Omega_q^T\right)} \\
&= \sum_{n,p} \gamma_{np} \nabla_\theta \log \mathcal{N}\left(x_n | \beta_p, \Omega_p \Omega_p^T\right),
\end{aligned}$$

with

$$\gamma_{np} = \frac{\pi_p \mathcal{N}\left(x_n | \beta_p, \Omega_p \Omega_p^T\right)}{\sum_{q \in \Phi} \pi_q \mathcal{N}\left(x_n | \beta_q, \Omega_q \Omega_q^T\right)},$$

the equation for the responsibilities. Although $\gamma_{np}$ generally depend on the parameter $\theta$, in the EM algorithm the responsibilities are assumed to remain constant when optimizing the model parameters in the M-step.

The E-step is very similar to that of a standard GMM, but instead of computing the responsibilities $\gamma_{nk}$ for every component $k$, one needs to compute them for every path $p = (p_1, p_2, \ldots, p_k) \in \Phi$. This is because every path represents a Gaussian mixture component in the equivalent shallow GMM. Because $\gamma_{np}$ needs to be computed for each datapoint independently, the E-step is very easy to parallelize. Often a simple way to increase the speed of convergence and to reduce computation time is to use an EM-variant with "hard" assignments. Here only one of the responsibilities of each datapoint is set to 1:

$$\gamma_{np} = \begin{cases} 1 & p = \arg\max_q \left(\pi_q \mathcal{N}\left(x_n | \beta_q, \Omega_q \Omega_q^T\right)\right) \\ 0 & \text{otherwise} \end{cases} \tag{4}$$

**Heuristic**

Because the number of paths is the product of the number of components per layer ($\prod_j^k N_j$), computing the responsibilities can become intractable for big Deep GMM networks. However, when using hard-EM variant (eq. 4), this problem reduces to finding the best path for each datapoint, for which we can use efficient heuristics. Here we introduce such a heuristic that does not hurt the performance significantly, while allowing us to train much larger networks.

We optimize the path $p = (p_1, p_2, \ldots, p_k)$, which is a multivariate discrete variable, with a coordinate ascent algorithm. This means we change the parameters $p_i$ layer per layer, while keeping the

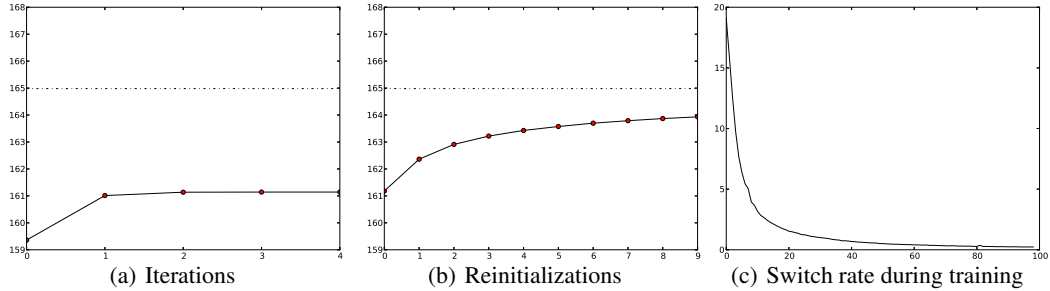

|  | (a) Iterations | (b) Reinitializations | (c) Switch rate during training |

Figure 2: Visualizations for the introduced E-step heuristic. (a): The average log-likelihood of the best-path search with the heuristic as a function of the number of iterations (passes) and (b): as a function of the number of repeats with a different initialization. Plot (c) shows the percentage of data points that switch to a better path found with a different initialization as a function of the number of the EM-iterations during training.

parameter values of the other layers constant. After we have changed all the variables one time (one *pass*), we can repeat.

The heuristic described above only requires $\sum_j^k N_j$ path evaluations per pass. In Figure 2 we compare the heuristic with the full search. On the left we see that after 3 passes the heuristic converges to a local optimum. In the middle we see that when repeating the heuristic algorithm a couple of times with different random initializations, and keeping the best path after each iteration, the log-likelihood converges to the optimum.

In our experiments we initialized the heuristic with the optimal path from the previous E-step (warm start) and performed the heuristic algorithm for 1 pass. Subsequently we ran the algorithm for a second time with a random initialization for two passes for the possibility of finding a better optimum for each datapoint. Each E-step thus required $3\left(\sum_j^k N_j\right)$ path evaluations. In Figure 2(c) we show an example of the percentage of data points (called the *switch-rate*) that had a better optimum with this second initialization for each EM-iteration. We can see from this Figure that the switch-rate quickly becomes very small, which means that using the responsibilities from the previous E-step is an efficient initialization for the current one. Although the number of path evaluations with the heuristic is substantially smaller than with the full search, we saw in our experiments that the performance of the resulting trained Deep GMMs was ultimately similar.

## 3.2 Maximization

In the maximization step, the parameters are updated to maximize the log likelihood of the data, given the responsibilities. Although standard optimization techniques for training deep networks can be used (such as SGD), Deep GMMs have some interesting properties that allow us to train them more efficiently. Because these properties are not obvious at first sight, we will derive the objective and gradient for the transformation matrices $A_{i,j}$ in a Deep GMM. After that we will discuss various ways for optimizing them. For convenience, the derivations in this section are based on the hard-EM variant and with omission of the bias-terms parameters. Equations without these simplifications can be obtained in a similar manner.

In the hard-EM variant, it is assumed that each datapoint in the dataset was generated by a path $p$, for which $\gamma_{n,p} = 1$. The likelihood of $x$ given the parameters of the transformations on this path is

$$p(x) = \left|A_{1,p_1}^{-1}\right| \dots \left|A_{k,p_k}^{-1}\right| \mathcal{N}\left(A_{1,p_1}^{-1} \dots A_{k,p_k}^{-1} x | 0, I_n\right), \tag{5}$$

where we use $|\cdot|$ to denote the absolute value of the determinant. Now let's rewrite:

$$z = A_{i+1,p_{i+1}}^{-1} \dots A_{k,p_k}^{-1} x \tag{6}$$

$$Q = A_{i,p_i}^{-1} \tag{7}$$

$$R_p = A_{1,p_1}^{-1} \dots A_{i-1,p_{i-1}}^{-1}, \tag{8}$$

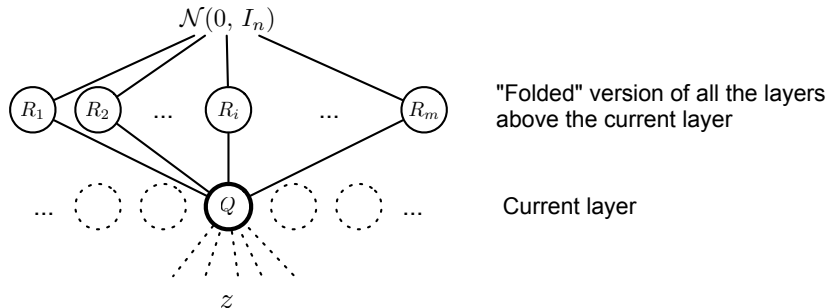

$\mathcal{N}(0,\ I_n)$

$R_1$ $R_2$ $\cdots$ $R_i$ $\cdots$ $R_m$

"Folded" version of all the layers above the current layer

$Q$

Current layer

$z$

Figure 3: Optimization of a transformation Q in a Deep GMM. We can rewrite all the possible paths in the above layers by "folding" them into one layer, which is convenient for deriving the objective and gradient equations of Q.

so that we get (omitting the constant term w.r.t. $Q$):

$$\log p\left(x\right) \propto \log|Q| + \log\mathcal{N}\left(R_pQz|0,\,I_n\right). \tag{9}$$

Figure 3 gives a visual overview. We have "folded" the layers above the current layer into one. This means that each path $p$ through the network above the current layer is equivalent to a transformation $R_p$ in the folded version. The transformation matrix for which we will derive the objective and gradient is called Q. The average log-likelihood of all the data points that are generated by paths that pass through $Q$ is:

$$\frac{1}{N}\sum_i \log p\left(x_i\right) \quad \propto \quad \log|Q| + \frac{1}{N}\sum_p\sum_{i\in\phi_p} \log\mathcal{N}\left(R_pQz_i|0,I\right) \tag{10}$$

$$= \quad \log|Q| - \frac{1}{2}\sum_p \pi_p Tr\left[\Gamma_p Q^T\Omega_p Q\right], \tag{11}$$

where $\pi_p = \frac{N_p}{N}, \Gamma_p = \frac{1}{N_p}\sum_{i\in\phi_p} z_i z_i^T$ and $\Omega_p = R_p^T R_p$. For the gradient we get:

$$\frac{1}{N}\nabla_Q\sum_i \log p\left(x_i\right) = Q^{-T} - \sum_p \pi_p\Gamma_p Q^T\Omega_p. \tag{12}$$

**Optimization**

Notice how in Equation 11 the summation over the data points has been converted to a summation over covariance matrices: one for each path[1]. If the number of paths is small enough, this means we can use full gradient updates instead of mini-batched updates (e.g. SGD). The computation of the covariance matrices is fairly efficient and can be done in parallel. This formulation also allows us to use more advanced optimization methods, such as LBFGS-B [12].

In the setup described above, we need to keep the transformation $R_p$ constant while optimizing $Q$. This is why in each M-step the Deep GMM is optimized layer-wise from top to bottom, updating one layer at a time. It is possible to go over this process multiple times for each M-step. Important to note is that this way the optimization of $Q$ does not depend on any other parameters in the same layer. So for each layer, the optimization of the different nodes can be done in parallel on multiple cores or machines. Moreover, nodes in the same layer do not share data points when using the EM-variant with hard-assignments. Another advantage is that this method is easy to control, as there are no learning rates or other optimization parameters to be tuned, when using L-BFGS-B "out of the box". A disadvantage is that one needs to sum over all possible paths above the current node in the gradient computation. For deeper networks, this may become problematic when optimizing the lower-level nodes.

Alternatively, one can also evaluate (11) using Kronecker products as

$$\cdots = \log|Q| + \mathrm{vec}\left(Q\right)^T\left\{\sum_p \pi_p\left(\Omega_p\otimes\Gamma_p\right)\right\}\mathrm{vec}\left(Q\right) \tag{13}$$

and Equation 12 as

$$\cdots = Q^{-T} + 2 \operatorname{mat}\left(\left\{\sum_p \pi_p \left(\Omega_p \otimes \Gamma_p\right)\right\} \operatorname{vec}(Q)\right). \tag{14}$$

Here $\operatorname{vec}$ is the vectorization operator and $\operatorname{mat}$ its inverse. With these formulations we don't have to loop over the number of paths anymore during the optimization. This makes the inner optimization with LBFGS-B even faster. We only have to construct $\sum_p \pi_p \left(\Omega_p \otimes \Gamma_p\right)$ once, which is also easy to parallelize. These equation thus allow us to train even bigger Deep GMM architectures. A disadvantage, however, is that it requires the dimensionality of the data to be small enough to efficiently construct the Kronecker products.

When the aforementioned formulations are intractable because there are too number layers in the Deep GMM and the data dimensionality is to high, we can also optimize the parameters using backpropagation with a minibatch algorithm, such as Stochastic Gradient Descent (SGD). This approach works for much deeper networks, because we don't need to sum over the number of paths. From Equation 9 we see that this is basically the same as minimizing the L2 norm of $R_p Q z$, with $\log |Q|$ as regularization term. Disadvantages include the use of learning rates and other parameters such as momentum, which requires more engineering and fine-tuning.

The most naive way is to optimize the deep GMM with SGD is by simultaneously optimizing all parameters, as is common in neural networks. When doing this it is important that the parameters of all nodes are converged enough in each M-step, otherwise nodes that are not optimized enough may have very low responsibilities in the following E-step(s). This results in whole parts of the network becoming unused, which is the equivalent of empty clusters during GMM or k-means training. An alternative way of using SGD is again by optimizing the Deep GMM layer by layer. This has the advantage that we have more control over the optimization, which prevents the aforementioned problem of unused paths. But more importantly, we can now again parallelize over the number of nodes per layer.

## 4    Experiments and Results

For our experiments we used the Berkeley Segmentation Dataset (BSDS300) [13], which is a commonly used benchmark for density modeling of image patches and the tiny images dataset [14]. For BSDS300 we follow the same setup of Uria et al. [15], which is best practice for this dataset. 8 by 8 grayscale patches are drawn from images of the dataset. The train and test sets consists of 200 and 100 images respectively. Because each pixel is quantized, it can only contain integer values between 0 and 255. To make the integer pixel values continuous, uniform noise (between 0 and 1) is added. Afterwards, the images are divided by 256 so that the pixel values lie in the range [0, 1]. Next, the patches are preprocessed by removing the mean pixel value of every image patch. Because this reduces the implicit dimensionality of the data, the last pixel value is removed. This results in the data points having 63 dimensions. For the tiny images dataset we rescale the images to 8 by 8 and then follow the same setup. This way we also have low resolution image data to evaluate on.

In all the experiments described in this section, we used the following setup for training Deep GMMs. We used the hard-EM variant, with the aforementioned heuristic in the E-step. For each M-step we used LBFGS-B for 1000 iterations by using equations (13) and (14) for the objective and gradient. The total number of iterations we used for EM was fixed to 100, although fewer iterations were usually sufficient. The only hyperparameters were the number of components for each layer, which were optimized on a validation set.

Because GMMs are in theory able to represent the same probability density functions as a Deep GMM, we first need to assess wether using multiple layers with a deep GMM improves performance. The results of a GMM (one layer) and Deep GMMs with two or three layers are given in 4(a). As we increase the complexity and number of parameters of the model by changing the number of components in the top layer, a plateau is reached and the models ultimately start overfitting. For the deep GMMs, the number of components in the other layers was kept constant (5 components). The Deep GMMs seem to generalize better. Although they have a similar number of parameters, they are able to model more complex relationships, without overfitting. We also tried this experiment on a more difficult dataset by using highly downscaled images from the tiny images dataset, see Figure

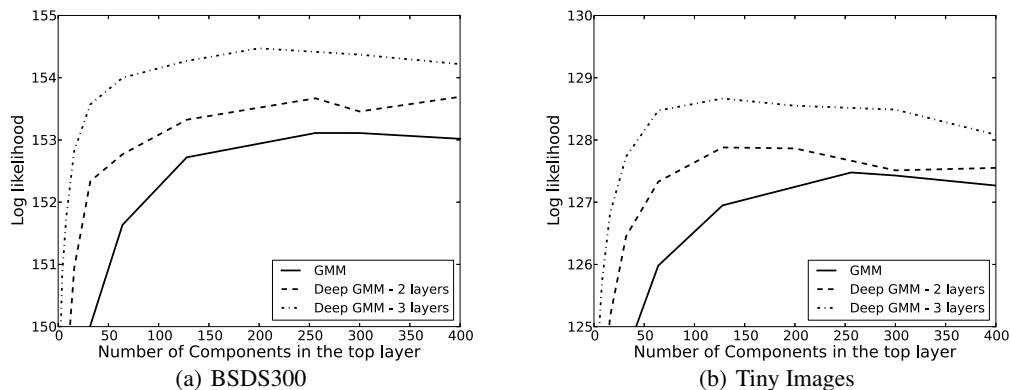

|  (a) BSDS300 |  (b) Tiny Images |
|---|---|

Figure 4: Performance of the Deep GMM for different number of layers, and the GMM (one layer). All models were trained on the same dataset of 500 Thousand examples. For comparison we varied the number of components in the top layer.

4(b). Because there are less correlations between the pixels of a downscaled image than between those of an image patch, the average log likelihood values are lower. Overall we can see that the Deep GMM performs well on both low and high resolution natural images.

Next we will compare the deep GMM with other published methods on this task. Results are shown in Table 1. The first method is the RNADE model, a new deep density estimation technique which is an extension of the NADE model for real valued data [16, 15]. EoRNADE, which stands for ensemble of RNADE models, is currently the state of the art. We also report the log-likelihood results of two mixture models: the GMM and the Student-T Mixture model, from [2]. Overall we see that the Deep GMM has a strong performance. It scores better than other single models (RNADE, STM), but not as well as the ensemble of RNADE models.

| Model | Average log likelihood |
|---|---|
| RNADE: 1hl, 2hl, 3hl; 4hl, 5hl, 6hl | 143.2, 149.2, 152.0, 153.6, 154.7, 155.2 |
| EoRNADE (6hl) | 157.0 |
| GMM | 153.7 |
| STM | 155.3 |
| **Deep GMM - 3 layers** | **156.2** |

Table 1: Density estimation results on image patch modeling using the BSDS300 dataset. Higher log-likelihood values are better. "hl" stands for the number of hidden layers in the RNADE models.

## 5   Conclusion

In this work we introduced the deep Gaussian Mixture Model: a novel density estimation technique for modeling real valued data. we show that the Deep GMM is on par with the current state of the art in image patch modeling, and surpasses other mixture models. We conclude that the Deep GMM is a viable and scalable alternative for unsupervised learning. The deep GMM tackles unsupervised learning from a different angle than other recent deep unsupervised learning techniques [17, 18, 19], which makes it very interesting for future research.

In follow-up work, we would like to make Deep GMMs suitable for larger images and other high-dimensional data. Locally connected filters, such as convolutions would be useful for this. We would also like to extend our method to modeling discrete data. Deep GMMs are currently only designed for continuous real-valued data, but our approach of reparametrizing the model into layers of successive transformations can also be applied to other types of mixture distributions. We would also like to compare this extension to other discrete density estimators such as Restricted Boltzmann Machines, Deep Belief Networks and the NADE model [15].

## Footnotes

[1]Actually we only need to sum over the number of possible transformations $R_p$ above the node $Q$.

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
