[Reviews · NeurIPS 2014]

Submitted by Assigned_Reviewer_21

This paper introduces the idea of deep Gaussian mixture models. A GMM can be seen as consisting of a single isotropic unit norm Gaussian, where each of the components of the mixture consists of applying a different linear transformation to that Gaussian. This idea is extended to the case of a multilayer network, where each node in the network corresponds to a linear transformation, and each route through the network corresponds to a sequence of linear transformations. The number of mixture components is then the number of routes through the network.

This is a clearly written paper. The idea is both simple and good. Overall I liked the paper. I did think that the experimental results were disappointingly low dimensional, especially considering how much effort was spent in the paper discussing how to make the algorithm computationally tractable.

Also, I think the proposed algorithm is quite closely related to
Tang, Yichuan, Ruslan Salakhutdinov, and Geoffrey Hinton. "Deep mixtures of factor analysers." International Conference on Machine Learning (2012).
this probably deserves a citation and discussion.

I would strongly encourage the authors to release the source code for the experimental results as supplemental material. This makes the paper more convincing, increases citation counts, and encourages a culture of reproducible science which benefits everyone.

More detailed comments follow:

39 - "in function" -> "as a function"

73 - "A_3,1" -> "A_1,3"

89 - You provide good arguments in 312-318 for why this is true for your algorithm. I'm skeptical that EM is more parallelizable than stochastic gradient descent in general though. Maybe phrase this more specifically.

139 - There are an exponential number of paths through the network. Assigning a probability to each of them without factorizing them would become intractable quite quickly -- even for relatively small networks. I'm surprised this factorization isn't needed almost all the time.

199 - This is commonly called the MAP approximation. Why not instead just sample a single nonzero gamma with probability proportional to pi (this is EM, with the E step represented as a sample from the posterior). Unlike when using the MAP approximation, this will give you an unbiased estimate of your learning gradient, and should lead to a higher log likelihood model.

230 - "in function of" -> "as a function of"

235 - In the plot it never converges to the optimum.

251 - "and scalable." -> "and in a scalable way."

307, 341 - If you want to do quasi-Newton optimization with minibatches, and without hyperparameters, you might try https://github.com/Sohl-Dickstein/Sum-of-Functions-Optimizer

312-318 - Cool. This provides good support for the benefits of EM.

331 - "have to construction" -> "have to construct"

337 - with using -> using

344 - I strongly suspect this is a side effect of the MAP approximation to the posterior, as opposed to not optimizing fully during the M step. (As a note -- EM can be seen as maximizing a lower bound on the log likelihood. Importantly, that lower bound increases even if the M step is not run to convergence.)

Figure 4 - I don't think this X-axis makes much sense for the Deep GMMs -- there's nothing special about the number of components in the top layer. Number of parameters would probably make a better x-axis.

436 - Author name listed as "anonymous".
Summary: The idea is simple, good, and clearly presented. Overall I liked the paper. I thought the experimental results were unexpectedly weak.

Submitted by Assigned_Reviewer_30

The paper proposes a Deep Gaussian Mixture model (Deep GMM), which generalizes Gaussian mixtures to multiple layers. The key idea is to stack multiple GMM layers on top of each other. One can view Deep GMMs as a generative model where a standard normal random variable is successively transformed through a path in a network of k layers, where a transformation (multiplication by a matrix and adding a bias) is performed at each layer of the network.

One can then construct an equivalent shallow GMM but with the exponential number of mixture components.

In general, this is quite an interesting idea and the authors provide various heuristics to speed up EM learning algorithm, including (1) using hard EM and (2) using a "folding" trick by folding all the layers above a current layer into a flat "shallow" GMM model (although this becomes expensive when considering bottom layers).

However, my main concern is that this work is very closely related to the following work on a deep mixture of factor analyzers:

Deep Mixtures of Factor Analysers (ICML 2012)
Yichuan Tang, Ruslan Salakhutdinov and Geoffrey Hinton

especially given the close connections between GMMs and mixture of factor analyzers. Similar to your construction, deep MFA can be "folded" into a shallow MFA and learning can be carried out using EM. One can also pretrain these models layer-by-layer.

I think it would be important to highlight similarities/differences between
your work and deep MFA work.
Summary: In general, this is a well-written paper. But given its similarity to the previously publish work, the authors need to clarify their novel contributions.

Submitted by Assigned_Reviewer_46

The paper presents a new model called DeepGMM, which is a MoG with an exponential number of components and tied parameters. Each component in the mixture correspond to a path through a fully interconnected multilayer network of affine transformations (where the matrix is the Cholesky of a positive-definite matrix)applied on a identity-covariance multivariate Gaussian. The weights of each component are not tied, although a factorisation is suggested in the paper for very deep networks.

As the authors comment, their hypothesis is that by tying the parameters of the many components a bigger number of components can be used while avoiding overfitting.

The authors also propose a hard-EM based training algorithm. For the expectation phase, coordinate descent and several heuristics are recommended to decrease the computational load. For the maximization phase the authors present three options:
A batch GD method suitable for DeepGMMs with a small number of paths. Unfortunately, the authors give no specific figures.
A batch GD method suitable for DeepGMMs modelling data of sufficiently small dimensionality
A SGD method for bigger DeepGMMs

Experimental results on two well-known dataset are presented. All experiments use the second of the aforementioned optimisation techniques.

Quality:
Pros: The paper is technically sound and it main hypothesis supported by experimental results. Figure 4 shows by tying parameters it is possible to train a DeepMoG with 2500 effective components that offers superior performance than a untied MoG using 300 (which has more parameters), and using more components in an untied mixture offers no improvement.

Cons: The paper only show results on natural images. The inductive bias of a DeepGMM could be specially advantageous on this kind of data.

Clarity
The paper is well structured and reads well, but there are some typos (parrallizable -> parallelizable, netwerk -> network).
In Figure 4 the maximum value achieved is about 154.5 while in Table 1 it is reported as 156.2. Are these different experiments using more data instead of 500 thousand patches? If so it should be specified.

It would be of interest to report training times, does it take hours, days or weeks to train a model of 8x8 patches?

In Figure 4, if the goal is to show, the ability to train a MoG with many components without overfitting, it would be more interesting to show the effective number of components instead of the number of components in the top layer.

If the authors find space it could be interesting to show some samples from the model.

Was the tinyimages dataset resized to 8x8 pixels, if not the likelihods should not be compared to those obtained on BSDS300 as is done on line 397.

Originality:
The particular technique for parameter tying in a GMM presented in this paper is new. Also the training algorithms presented (including heuristics) are of interest.

Significance:
The results are important. Although training and evaluation of densities at test time will have high computational cost, sampling should be very efficient. Also the idea is interesting and can be further built upon.
Summary: The paper presents a new way of tying the parameters of a MoG that allows the authors to obtain state-of-the-art results on patches of natural images. The paper is interesting and easy to read.
Author Feedback
Author rebuttal: Assigned_Reviewer_21:

I'll add the "Deep mixtures of factor analysers" citation + discussion to the paper.

I'm working towards making the source code available.

I agree with most of the detailed comments you made. I'll make appropriate changes where necessary.
Thanks for the useful suggestions (Especially your comment at 199 is interesting. I'll take a closer look.)

Assigned_Reviewer_21:

I'll make sure to make the distinctions and similarities with the Deep MFA technique clear in the camera ready version.

These techniques are actually more different than their names would suggest:
The nodes in a Deep mixture of factor analyzers do not share the same lower-level nodes (a tree structure), while in the deep GMM all nodes are connected to all the nodes in the layer below (Directed acyclic graph). In the DMFA paper, training is performed layer by layer, where the dataset is hierarchically clustered and the children of each node are trained as a MFA on a different subset of the data using the standard EM algorithm. This means the parameters of the parent nodes are not updated when training its children. With the DGMM the different layers are adapted to each other and the parameters are jointly optimizing the log-likelihood objective. Finally, the parameter sharing in the proposed model is also constructed differently.

Assigned_Reviewer_46:

Thank you for the useful comments and suggestions.
I'll make appropriate changes where necessary.